# On the Reliability of Temperature Measurements in Natural Gas Pipelines

**DOI:** 10.3390/s23063121

**Published:** 2023-03-14

**Authors:** Giorgio Ficco, Marialuisa Cassano, Gino Cortellessa, Fabrizio Zuena, Marco Dell’Isola

**Affiliations:** 1Department of Civil and Mechanical Engineering (DICEM), University of Cassino and Southern Lazio, 03043 Cassino, Italy; 2Snam S.p.A., 20097 San Donato Milanese, Italy

**Keywords:** natural gas, unaccounted for gas (UAG), transmission network, thermowell, closed conduits

## Abstract

Accuracy of temperature measurement of natural gas flows in closed conduits is a highly debated topic due to the complexity of the measurement chain and the related economic impact. First, specific thermo-fluid dynamic issues occur because of the difference between the temperature of the gas stream and that of the external ambient and the mean radiant temperature inside the pipe. Furthermore, the installation conditions of the temperature sensor (e.g., immersion length and diameter of the thermowell) play a crucial role. In this paper, the authors present the results of a numerical and experimental study conducted both in the laboratory and in-field aimed at analyzing the reliability of temperature measurement in natural gas networks as a function of the pipe temperature and of the pressure and velocity of the gas stream. The results obtained in the laboratory show errors ranging between 0.16 and 5.87 °C in the summer regime and between −0.11 and −2.72 °C in the winter regime, depending on the external pipe temperature and gas velocity. These errors have been found to be consistent with those measured in-field, where high correlation between the pipe temperatures, the gas stream and the external ambient have been also demonstrated, especially in summer conditions.

## 1. Introduction

Reliability of gas temperature measurement in a closed conduit is crucial for the management of transmission and distribution natural gas networks because of the potential consequences for accuracy of consumption and network balancing. As is well known, to convert the measured gas volumes at operative conditions into standard references, accurate knowledge of both the thermodynamic conditions of the gas (i.e., temperature and pressure) and the chemical composition of the gas is necessary [1,2]. Therefore, a complex measuring chain is needed for natural gas measurements, and in [3], the typical uncertainty budget of the measuring plants in the transmission network was investigated. 

However, measuring the temperature of a fluid flowing in high pressure pipelines, both in stationary and dynamic conditions, presents numerous critical issues arising from the complex energy exchanges between the probe and the surrounding environment. In particular, the temperature probe for gaseous flows in closed conduits exchanges energy with the gas, as well as with the internal surface of the pipe and the external environment. Therefore, even small errors in measuring the temperature of the gas stream (e.g., ones deriving from the difference between the ambient temperature and that of the gas) can lead to non-negligible errors in accounting gas volumes, especially in the case of large flow meters. As a consequence, systematic errors strongly affect the physical and commercial balancing of the network and influence the related unaccounted for gas (UAG). 

The effects of temperature measurement accuracy of gas flows in pipelines have been investigated both in the transmission [4] and distribution networks [5,6], demonstrating that a clear correlation occurs between temperature measurement accuracy and UAG, which is a very crucial issue in modern networks. In particular, in [7], the practical advantages gained by implementing statistical monitoring of the UAG time series and focusing on the presence of seasonality are highlighted in the UK transmission network for different UAG sources.

Resistance temperature detectors (RTD) with 4-wire connection and platinum sensor (e.g., Pt100), typically enclosed in a relatively thin stainless steel sheath (approx. 5–6 mm), are commonly used for gas temperature measurement in transmission networks [8]. These sensors are generally installed in a thermowell and aim to protect the sensor from the flow itself and allow easy management for maintenance and verification [9]. The related installation effects, however, make the measurement more complex since direct contact of the temperature probe with the gas is avoided, thus influencing both the response time and the measurement accuracy [10,11]. Standard EN ISO 15970:2014 [12] sets an immersion length of approximately 1/3 of the nominal internal diameter of the pipe. Furthermore, the immersion length should not exceed 3/4 of the nominal pipe diameter; if it does, a 45° installation or an internal pipe bend is be preferred. Additionally, [13] investigates the optimal location of temperature sensors in pipe flows that are used to determine the gas temperature in flow metering applications. The thermowell is normally mounted orthogonally to the flow to maximize the convective heat exchange. However, in the measurement model, radiative heat exchanges between the thermowell and the pipe walls also occur concurrently with conductive exchanges through the stem of the thermowell. As a secondary effect, which is generally negligible, the convective heat exchange between the sensor and the external environment should be considered. In fact, under both steady and dynamic conditions, the temperature measured by the probe can be significantly different from that of the gas stream. Thermo-fluid dynamics issues related to the installation of the coupling sensor-thermowell will also strongly affect the accuracy and reliability of measurements [14,15]. Furthermore, a temperature gradient along the axis of the pipe can lead to a temperature difference between the section of the thermowell and the section where the flow meter is installed. In this regard, technical standards generally establish a minimum distance of two nominal diameters (DN) for volumetric meters [16]. In [17], it was demonstrated that this issue can lead to errors in volume of up to 0.06%.

Theoretically, the surface emissivity of the thermowell should be low enough to reduce radiative exchange between the sensor and the internal surface of the pipe. Even though the wells used in natural gas applications are typically made of polished stainless steel, the real emissivity, especially in metals, can vary considerably as a function of the degree of surface finish, the oxidation, and the presence of contaminants [18]. Therefore, since the real wear of the thermowells in operation is not easily predictable, this contribution cannot be neglected, especially when high temperature differences occur between the pipe and the sensor (i.e., during the summer regime). 

With the aim of developing a useful tool for the estimation of measurement errors due to conduction along the stem and radiative exchange between the pipe wall and the well, in [19], an experimental comparison was carried out in a pipeline with natural gas at the *P* = 60 bar in an environmental control box where the temperature was set to vary within a range from 20–50 °C (i.e., summer regime). The results showed a decrease in the measurement error as the gas velocity increased and the outside air temperature decreased. Furthermore, since a maximum deviation of about −6% was found between the experimental measurements and the proposed mathematical model, it is possible to adopt the latter to evaluate the accuracy of the measurement in summer conditions. Finally, specific studies have been conducted recently that examine how the installation conditions influence the accuracy of the temperature measurement in the transmission and distribution natural gas networks. These studies have considered flanged thermowells by using both a CFD analysis [20] and an experimental campaign performed on 3″, 8″ and 24″ pipes with gas velocities ranging between 0.1 to 14 m/s [21]. In particular, in [20], the influence of the external environment on the temperature of the top of the thermowell in summer conditions (where the influence is higher when the ambient temperature increases) as well as in winter conditions (where the influence is higher as the ambient temperature decreases) has been demonstrated through a numerical investigation. The authors concluded by highlighting the need for thermal insulation of the measurement stretch. Conversely, the experimental study in [21] demonstrated that insulation of the pipes is not necessary and proposed an alternative method to measure the gas temperature through contact sensors on the pipe surface. It has been found pipe surface temperature measurements generally show errors within 0.5 °C for gas velocities higher than 1 m/s and ambient temperature in the range from −20 to +20 °C. More recently, non-invasive time-dependent temperature measurements in pipeline applications have been widely proposed to measure the temperature of fluids in pipes. The model proposed in [22] allows for the prediction of temperature evolution inside the pipe for both convective and conductive thermal exchanges. However, the main problem with non-invasive methods is that the temperature measured on the external pipe surface is always lagged in respect to that of the gas stream due to the thermal diffusion phenomena. As an example, in [23], a non-invasive thermometric system based on the dual heat flux method was presented as a possible alternative to the use of thermowells. The authors claimed that their method showed measurement accuracy within 1 °C, thus obtaining performances comparable to those of thermowells. In [24], numerical simulations were performed to quantify the differences between the measured temperature on the outside surface of a pipe and the temperature of the fluid flowing in the pipe. A correlation algorithm was also proposed to correct deviations between the measured outside pipe surface temperature and the fluid temperature. In [25], a method that aimed to determine heat transfer coefficients on the inner surface of the pipeline and outer surface of a thermowell immersed in a fluid which was flowing under high pressure was proposed. The immersion length and the dimensions (e.g., the diameter) of the stem also enhanced conduction phenomena along the axis of the thermowell, thus influencing the measurement accuracy. In [26], a theoretical model capable of quantifying the temperature measurement error associated with the conduction exchange along the wires of a thermocouple was discussed and experimentally validated. Finally, in [27], the development and validation of a theoretical model reproducing the installation effects of temperature probes in closed conduits was presented. The authors investigated different operative conditions of natural gas transmission networks and theoretically estimated typical random and systematic errors. Finally, they proposed some installation and operative solutions aimed at mitigating these installation effects. 

In this paper, the authors present and discuss the results of an experimental campaign aimed at estimating reliability of gas temperature measurements in closed conduits, both at laboratory conditions and in-field. First, experimental measurements were conducted in the laboratory at ambient pressure. Subsequently, aiming at reproducing the real operative conditions in the pipe, the authors developed a numerical model, based on the finite difference method, which was capable of scaling-up the measured errors at the higher line pressures typical of the natural gas transmission networks. This numerical model was validated against the laboratory measurements and then adopted for the estimation of measurement errors at different thermo-fluid dynamic and geometric conditions. Finally, an experimental campaign was conducted in-field in a regulating and measuring plant at the interconnection between the transmission and distribution networks. The in-field campaign allowed gas temperature measurement errors to be investigated at actual operative conditions, highlighting the thermo-fluid-dynamic differences compared to those performed in the laboratory. In both cases, the authors investigated the winter (i.e., low ambient temperature and high flow rate) and summer (i.e., high ambient temperature and low flow rate) regimes. The main novelty of this study consists in how the criticalities of temperature measurements of the natural gas stream in pipelines are investigated in laboratory and in-field from both a theoretical and experimental point of view. The obtained results could be useful: (i) to technicians involved in temperature measurements of working gases in industries and networks and (ii) to stakeholders in the field of natural gas (e.g., transmission and distribution operators, legal metrology authorities, etc.). At the same time, the main limitation of this study involves the investigation of a limited velocity field of the gas stream and boundary conditions (i.e., the typical operative conditions of the measuring plant investigated in-field). This implies that some of the measured deviations could be slightly different due to the different weight of the heat exchanges between the pipe and the sensors (i.e., radiative, conductive, and convective) and the effects of further influence from quantities that are virtually present in-field (such as installation effects).

## 2. Theory and Methods

As is well known, to account for consumption and to balance networks, natural gas volumes measured at operative conditions must be converted into standard reference measurements (i.e., in Italy, 288.15 K and 1.01325 bar). Therefore, a volume conversion device is associated with the flow sensor, usually consisting of a flow computer (i.e., electronic data acquisition and processing device), a pressure transmitter and a temperature transmitter. The volume $V_{S}$ at standard reference conditions is thus given by the following equation:(1)$$V_{S}=V\cdot KTvo=V\cdot \frac{P}{P_{S}}\cdot \frac{T_{S}}{T}\cdot \frac{Z_{S}}{Z}$$
where KTvo is the coefficient for the volumetric conversion, dimensionless; V ($V_{s}$) is the volume at operative (standard reference) conditions, m^3^; P ($P_{S}$) is the absolute pressure at operative (standard reference) conditions, bar; T ($T_{S}$) is the absolute temperature at operative (standard reference) conditions, K; Z ($Z_{S}$) is the compressibility factor at operative (standard reference) conditions, dimensionless.

In transmission networks, the gas coming from the underground pipelines is measured after passing through a section exposed to the external environment and this can lead to significant errors in measuring the gas temperature, which is influenced by several factors such as:the presence of preheating systems to cope with the Joule–Thomson effect (i.e., the lowering of the temperature following a gas pressure reduction);the absence of a cabin protecting the measurement stretch from the external environment;the absence of insulation for the measurement stretch.

Measurement errors due to the conductive and radiative exchange on the thermowell derive from the temperature difference between the natural gas and the pipe. This difference is raised in regulation and measurement plants where the pipe is brought above ground, and even the solar radiation plays a significant role. Aiming to lower the measurement error, it is therefore necessary to evaluate and possibly mitigate the following contributions: (a) the conductive thermal flow between the sensor and the top of the thermowell placed on the wall of the pipe; (b) the radiative heat flow between the thermowell and the pipe; (c) the thermal profile distortions; and (d) the temperature difference between the flow measurement section and that of the well.

The choice of the probe–well assembly, the position and connection of the well and, finally, its length and shape are crucial for accurate measurements to be obtained. Regarding the probe-well assembly, it is essential to ensure good thermal contact between the sensor and the thermowell. In particular, it is necessary to ensure that the tip of the sensor is in contact with the internal bottom of the well. Some installation schemes include spring-loaded sensors, which help ensure the best contact in any installation condition and orientation. The probe–pocket radial clearance should also be as limited as possible since the air itself acts as a thermal insulator. The use of conductive pastes or oils can partially mitigate this issue.

### 2.1. The Experimental Campaign in the Laboratory

Aiming at experimentally estimating the temperature measurement error in high pressure natural gas pipelines, the authors designed and developed a specific experimental campaign at the LAMI, the accredited laboratory of the University of Cassino and Southern Lazio. 

A DN 160 pipe was used to reproduce in the laboratory the gas flow in a transmission network under different conditions in winter ($T_{set}$ = 8, 12 °C and w = 7 m s^−1^) and in summer ($T_{set}$ = 30, 40, 50 °C and w = 0.5 m s^−1^). The above mentioned gas temperatures and velocities were chosen considering the typical values of the plant investigated in-field, i.e., pipe temperature never below 8 °C (due to winter climate that is not particularly cold) and not very high flow velocities due to the plant characteristics (medium pressure pipeline serving a distribution city network).

To this aim, the pipe was immersed in a thermostatic bath to keep its surface temperature constant. The fluid flow in the duct was simulated by aspirating air at ambient pressure and different velocities. Three immersion RTD (for measuring the temperature in the thermowell and those at the pipe inlet and outlet) together with two contact RTD (for measuring the external and internal surface temperatures of the pipe) were installed. The schematic in Figure 1 describes the test layout, in which $T_{flow}$ is the reference temperature, i.e., the temperature measured directly in the gas stream, close to the thermowell tip, and shielded from the radiative influence of the pipe walls. After an adequate stabilization time and the achievement of a steady state inside the pipe, temperature data in the six positions indicated in Figure 1 were gathered.

### 2.2. Development of a Numerical Model for Estimating the Gas Temperature 

Conditions in the field are different from those reproduced in the laboratory in this experimental campaign, which should be emphasized. Therefore, to correctly evaluate the obtained experimental results, it is necessary to consider the pressure conditions typical of transmission networks. To this aim, a finite difference method (FDM) numerical model has been specifically developed by the authors for the estimation of $T_{well}$ at typical pressure conditions (e.g., at 5, 24 and 30 bar) of natural gas transmission networks [28]. In particular, the estimation of $T_{well}$ requires the following input parameters: (i) gas velocity, $w\;\left({m\;s}^{-1}\right)$, (ii) absolute gas pressure at operative conditions, *P* (bar), (iii) thermowell diameter $D_{t}$ (m) and length $L_{t}$ (m), (iv) temperature of the internal surface of the pipe, $T_{pipe,int}$ (K), (v) flow temperature measured with a shielded sensor inside the pipe, $T_{flow}$ (K), (vi) thermowell conductivity $\lambda_{t}$ (W m^−1^ K^−1^) and emissivity *ε* (-).

The developed numerical model describes the interaction between the thermowell, assumed to be a cylinder, and the surrounding environment involving the three heat transfer mechanisms (conduction, convection and radiation) at steady state conditions. The thermowell (1D domain) has been simulated through six nodes, which aim to obtaining a uniform computational grid of the temperature profile (Figure 2a). The energy balance applied to a single node was obtained through the following equation (Figure 2b):(2)$${\dot{Q}}_{\left(i-1\right)\left(i\right)}+{\dot{Q}}_{\left(i+1\right)\left(i\right)}+{\dot{Q}}_{\infty ,i}+{\dot{Q}}_{r,i}=0$$
where ${\dot{Q}}_{\left(i-1\right)\left(i\right)}$ and ${\dot{Q}}_{\left(i+1\right)\left(i\right)}$ represent the conductive heat flux inside the pipe (W) and ${\dot{Q}}_{\infty ,i}$ is the convective heat flux (W); ${\dot{Q}}_{r,i}$ is the radiative heat flux (W).

Explaining the terms of the thermal balance by using a Taylor series, the heat exchange equation is obtained as follows: (3)$$\frac{T_{i-1}-T_{i}}{R_{\left(i-1\right)\left(i\right)}}+\frac{T_{i+1}-T_{i}}{R_{\left(i+1\right)\left(i\right)}}+h_{c}C\;\delta x\;\left(T_{\infty }-T_{i}\right)+4\;C\;\delta x\;\sigma \;\varepsilon \;T_{char}^{3}\left(T_{pipe,int}-T_{i}\right)=0$$
where (i) $T_{i-1}$, $T_{i}$ e $T_{i+1}$ are the nodal temperatures (K); (ii) $R_{\left(i-1\right)\left(i\right)}$ and $R_{\left(i+1\right)\left(i\right)}$ are the conductive thermal resistances (K W^−1^); (iii) δx is the height of the one-dimensional element (m); (iv) $T_{pipe,int}$ is the internal pipe temperature (K); (v) C is the external circumference of the pipe (m); (vi) $T_{char}$ is a characteristic temperature which in the case of small variations between $T_{pipe,int}$ and $T_{i}$ can be assumed to be equal to their average (K); (vii) $h_{c}$ is the convective heat transfer coefficient (W/m^2^); (viii) σ is the proportionality factor in the Stefan–Boltzmann law (W m^−2^ K^−4^). Finally, a Dirichlet and a Robin boundary condition were applied at node 1 and node 6, respectively, for the resolution of the numerical model.

Aiming to resolve Equation (3), at gas velocity equal to 0.5 and 7 m s^−1^, the authors calculated $h_{c}$ by using the numerical correlations for forced convection [29] and neglecting, therefore, the free convection mechanism. Since the Richardson number, $R_{i}$, was lower than 0.1 in all the performed numerical analyses, this hypothesis was verified.

### 2.3. The Experimental In-Field Campaign

The experimental campaign was conducted in-field in a regulating and measuring plant at the interconnection between the national transmission network and the local distribution network. The investigated plant is located in the outskirts of Empoli (central Italy) and is made up of DN150 pipelines that operate at a regulated pressure of 24 bar. In this case, since natural gas consumption is almost exclusively used by the civil sector, the operation of space heating during winter creates substantially different regimes in the winter and summer. 

In the period from September 2019 to September 2020, data from the fiscal measuring system of the TSO (i.e., volume, pressure and temperature of the gas flow, $T_{well}$) have been analyzed, together with temperature data from sensors specifically installed on the external surface of the pipeline and in a thermowell mounted in an insulated portion of the pipe, $T_{well,\;ins}$, and close to the fiscal one. Hourly ambient temperature and solar radiation data for the specific experimental campaign site was also analyzed [30]. Unfortunately, since a shielded sensor installed directly in the gas stream in the proximity of the thermowell was not available due to regulatory constraints [31], the reference temperature of the gas stream, $T_{flow}$, was obtained through the model described in Section 2.2. In this case, since the developed model has been successfully validated in the range of the experimental runs, the calculated $T_{flow}$ could be considered reliable. Figure 3 presents a sketch of the sensors’ layout.

## 3. Results 

In this section, the results obtained during the in-laboratory campaign (*P* = 1 bar) are presented in terms of temperature (measured/estimated at the different points of the experimental apparatus as described in Section 2.1) and measurement errors. Subsequently, the results of the validation of the numerical model against the experimental data at ambient pressure are presented together with the errors predicted at different line pressures (5, 24, and 30 bar). The results of the in-field campaign are then reported, highlighting differences and similarities with in-laboratory data. 

### 3.1. In-Laboratory Campaign 

In Table 1 the experimental results of the in-laboratory campaign are reported together with the absolute ($E_{meas}=T_{well,meas}-T_{flow}$) and relative ($E{\%}_{meas}=E_{meas}/T_{flow}*100$) measured errors. 

As expected, data in Table 1 shows that $T_{flow}$, which is the temperature measured in the gas stream through a shielded sensor, is always between $T_{flow,in}$ and $T_{flow,out}$. Furthermore, $T_{well}$, which is the temperature measured in the thermowell, is always overestimated in the summer regime, with errors ranging from +1.9 °C (at $T_{set}=$ 30 °C and w = 7.0 m s^−1^) to +11.60 °C (at $T_{set}=$ 50 °C and w = 0.5 m s^−1^). In the winter regime, $T_{well}$ is always underestimated, with absolute errors ranging from −0.70 °C (at $T_{set}=$ 12 °C and w = 7.0 m s^−1^) to −4.2 °C (at $T_{set}=$ 8 °C and w = 0.5 m s^−1^). As expected, this latter effect is lower in respect to that in the summer regime and occurs to a greater extent at lower flow rates. 

Obviously, it is worthy to underline the difference between the conditions reproduced in the laboratory and those in-field in terms of fluid velocity and surface pipe temperature as well as operative line pressure. As far as the so-called city gate measuring plants operated by DSO are concerned, it should be noted that summer conditions are the most difficult since flow rates are significantly reduced and ambient temperatures are much higher. However, this effect should be even more attenuated in winter due to the higher flow rates related to the increased natural gas demand by the residential sector.

The developed numerical model was validated against experimental data at ambient pressure and at flow velocities equal to 0.5 and 7 m s^−1^. The results of the validation process are shown in Table 2 together with the estimation of $T_{well}$ and of the related absolute deviation ${∆}_{model}=T_{well,model}-T_{well,meas}$ at different line pressures (i.e., 5, 24 and 30 bar, typical of the transmission network). It is worthy to note that, in Table 2, the average values of three runs for each of the 10 experimental points have been reported. The relative errors between numerical and experimental data have been found within 4% for all the investigated conditions, and therefore show good agreement. 

The data in Table 2 shows that the absolute deviation between experimental measurements and numerical estimation: (i) ranges between 0.22% (at $T_{set}$ = 30 °C) and 1.16% (at $T_{set}$ = 8 °C) at low flow conditions, and (ii), at high flow conditions, is always within 0.18% regardless of the temperature. This effect is certainly attributable to the influence of the forced convection.

Finally, by applying the developed numerical model, the authors calculated the error at line pressure to be 5, 24, and 30 bar (see Table 2). As expected, it can be pointed out that this effect, although non-negligible, is strongly attenuated compared to that at ambient pressure. For example, when *P* = 30 bar, the estimated error varies from 0.16 °C (at $T_{set}$ = 30 °C and w = 7 m s^−1^) to 5.87 °C (at $T_{set}$ = 50 °C and w = 0.5 m s^−1^) in the summer regime. This leads the converted volume to be underestimated in the range between 0.05 and 1.95%. Conversely, the estimated error ranges between −0.11 °C (at $T_{set}$ = 12 °C and w = 7 m s^−1^) and −2.72 °C (at $T_{set}$ = 8 °C and w = 0.5 m s^−1^) in winter regime, leading to overestimation of the converted volume in the range 0.04−0.93%. Table 2 also shows that the estimated error decreases as the line pressure increases.

### 3.2. In-Field Campaign

In Figure 4a, the trend of the measured average daily temperature of the gas flow $T_{well}$ and of the ambient $T_{amb}$ have been reported together with the daily cumulated solar radiation. It can be highlighted that $T_{well}$ roughly ranges from 11.4–27.3 °C, which is clearly higher than the corresponding $T_{amb}$ in winter (see Figure 4c) and slightly lower in summer (see Figure 4b). Furthermore, the trend of $T_{well}$ is quite similar to the cumulated daily solar radiation trend, especially in the summer regime. 

As far as the difference of daily average $T_{pipe}-T_{well}$ is concerned, this latter ranges between −0.5 °C and +0.3 °C, respectively, in winter (i.e., January 2020, with measured average $T_{pipe}$ = 12.2 °C and w = 5.0 m s^−1^) and in summer (i.e., August 2020, with measured average $T_{pipe}$ = 24.0 °C and w = 0.5 m s^−1^). The significant correlation between $T_{pipe}$ and $T_{well}$ is evident in Figure 5a, which is a result of the combined effect of ambient temperature, solar radiation, and flow rates. In Figure 5b, the correlation between $T_{well}$ and the registered flow rate Q is depicted. It is worth noticing that the correlation is negligible in the winter regime, whereas in the summer regime, a clear negative correlation is found (R = −0.87). This highlights the influence of the convective effect of the flow rate: the lower the flow rate, the higher the measured temperature.

The results of the correlation analysis of the measured $T_{well}$ with the average daily ambient temperature $T_{amb}$ and the daily cumulated solar radiation $\sum G_{i}$ have been reported in graphs in Figure 5c,d. First, it can be pointed out that $T_{well}$ is not correlated with either $T_{amb}$ or $\sum G_{i}$ in the winter regime. However, in the summer regime, a non-negligible positive correlation of $T_{well}$ is demonstrated with both $T_{amb}$ (i.e., correlation coefficient R = 0.84) and $\sum G_{i}$ (i.e., R = 0.76). 

Finally, by applying the developed numerical model, the authors estimated the average error values to be $E_{meas}=$ 0.54 °C (0.18%) in the summer regime; $E_{meas}$ was negligible in the winter regime (i.e., within 0.03 °C and 0.01%). These values are quite consistent with those measured in the laboratory and reported at *P* = 24 bar (see Table 2). Such error, although unavoidable, could be mitigated (e.g., by insulating the measuring stretch of the pipeline); in any case, awareness of this error could assist in managing balancing issues of the network (e.g., UAG). It is worth noticing that the obtained results are consistent with those available in the literature [19].

## 4. Discussion

From the results of the experimental campaigns carried out in-laboratory and in-field and the related numeric analyses, it can be pointed out that, due to high radiative loads and low velocities (i.e., higher time duration), the radiative contribution of the pipe is larger in summer than in winter, leading to different extents of errors for the measured gas temperature. In particular, this error depends on the flow rate inside the pipe. The lower the flow rate, the higher the temperature error. This is particularly true in the summer regime due to the low flow rates commonly registered. Furthermore, the correlation of the measured temperature with both the external ambient temperature and the flow rate in summer confirms the combined effect of the radiation contribution from the pipe walls and of the convective effect related to the gas stream velocity. As expected, the line pressure also plays a relevant role; this contributes to reducing estimated errors at higher pressures compared to those at ambient pressure, but the remaining errors are still not negligible. 

A further relevant issue influencing the reliability of gas temperature measurement in closed conduits is represented by how the thermowell is often installed in a different position in respect to the flow meter. In this case, radiative (from the pipe) and convective (from the gas stream) contributions can cause variation in the measured gas temperature. This effect is substantially enhanced by the high pipe temperatures and low flow velocities that are typical of the summer regime.

According to Equation (1), the overestimation of the gas temperature measurements which tend to occur in summer determines the underestimation of gas volumes at standard reference conditions, leading to an increase in UAG. 

It is therefore understood that the reliability of temperature measurements is crucial to limit UAG in natural gas transmission networks. Therefore, in order to effectively reduce the error of the gas temperature measurements in pipelines, the following actions would be useful:to provide an effective insulation and/or shielding of the stretch and of the measurement sensors;to use suitable conductive coupling fluids in the well–probe contact;to adequate design the size (i.e., length and thickness) and inclination (e.g., oblique for small diameters) of the thermowells;to use shielded thermowells or finish them with low-emissivity surfaces;to adopt appropriate thermowell immersion lengths that conform to the applicable standards and manufacturer’s specifications;to reduce the distance between the sections of the pipe in which the gas flow rate and temperature are measured.

## 5. Conclusions

The measurement of natural gas temperature in closed conduits presents several critical issues, relating to the installation conditions of the sensor, the measurement methods, and the absence of insulation along the measuring stretch. Aiming to assess the reliability of temperature measurements of gas flows in pipelines, the authors developed a theoretical and experimental study that reproduced the winter and summer conditions in terms of typical temperatures and flow velocities.

The obtained results show that:in the laboratory at ambient pressure, the measured error varies: (i) from 1.88 °C (at $T_{set}$ = 30°C and *w* = 7 m s^−1^) to 11.60 °C (at $T_{set}$ = 50 °C and *w* = 0.5 m s^−1^) in the summer regime, (ii) from −0.70 °C (at $T_{set}$ = 12 °C and *w* = 7 m s^−1^) to −4.21 °C (at $T_{set}$ = 8 °C and *w* = 0.5 m s^−1^) in the winter regime; the measured error is greatly attenuated at higher line pressure (e.g., at 30 bar, the estimated error ranges between 0.16 °C and 5.87 °C and between −0.11 °C and −2.72 °C, for summer and winter regimes, respectively);the in-field campaign has shown relevant correlations in the summer between $T_{well}$ and: (i) the ambient temperature $T_{amb}$ (R = 0.84), (ii) the cumulated solar radiation $\sum G_{i}$ (R = 0.76), and (iii) the gas flow rate Q (R = −0.87);an average error value equal to 0.54 °C (0.18%) has been found in the field in summer, whereas in winter, this was negligible (i.e., within 0.03 °C and 0.01%).the measured errors in-field were found to be consistent with the corresponding ones in the laboratory.


The main finding of this research is that gas temperature in pipelines is systematically overestimated in summer regimes due to the combined effects of radiation from the pipe walls and convection from the gas stream. Under particular conditions (i.e., high ambient temperature and solar radiation together with low flow velocities), this effect could lead to underestimate measured gas volumes up to 0.5–1% and, consequently, to increases in UAG. However, due to the lower gas volumes transported in the summer regimes, this effect is attenuated in the yearly balance. Future development of this research could investigate the effect of temperature measurement errors in main transmission pipelines, i.e., ones at higher pressure (i.e., up to 60–70 bar) and flow velocities (i.e., up to 15–20 m s^−1^).

## Figures and Tables

**Figure 1 sensors-23-03121-f001:**
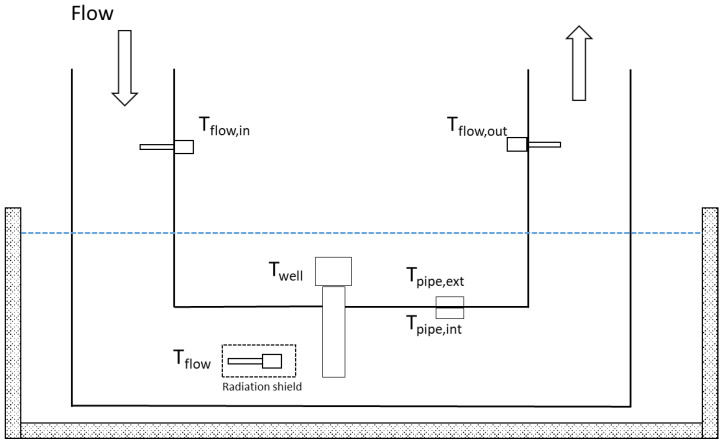
Sketch of the experimental layout for the in-laboratory measurement campaign.

**Figure 2 sensors-23-03121-f002:**
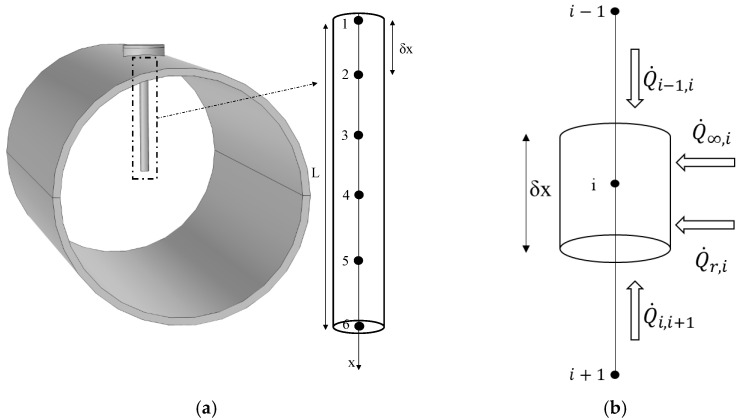
(**a**) Computational domain and grid, (**b**) Energy balance applied to a generic node.

**Figure 3 sensors-23-03121-f003:**
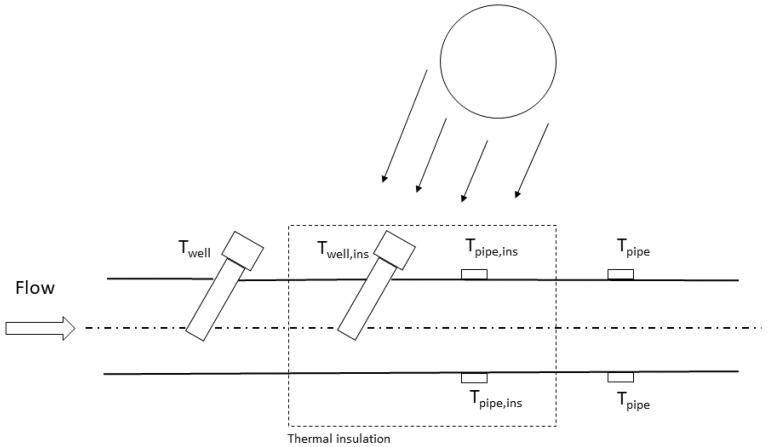
Sensors’ layout for the experimental campaign in-field.

**Figure 4 sensors-23-03121-f004:**
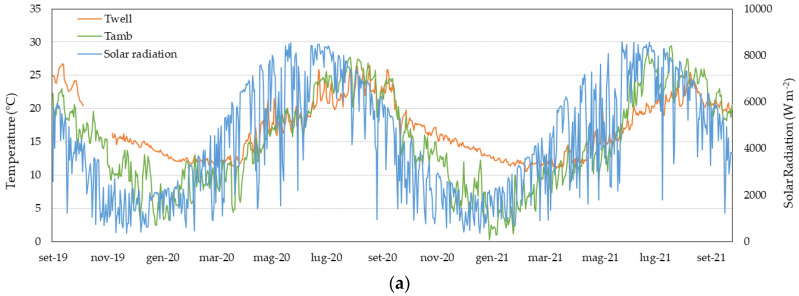
Experimental data trends of: (**a**) daily average ambient and gas temperature and daily cumulated solar radiation (secondary axis); daily average $T_{well}$, $T_{pipe}$, $T_{amb}$, and daily cumulated solar radiation (secondary axis) in winter (**b**) and summer (**c**) regimes.

**Figure 5 sensors-23-03121-f005:**
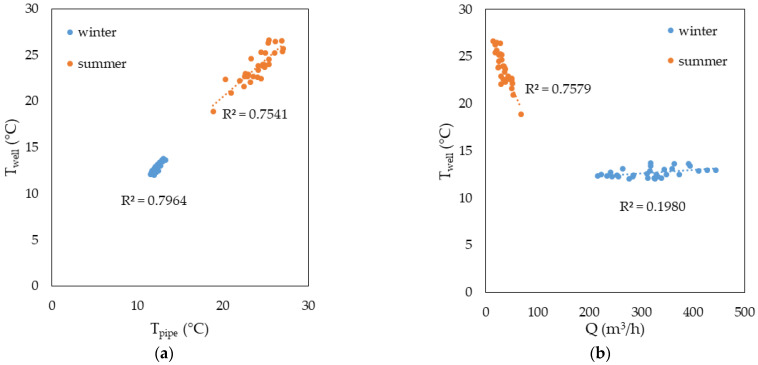
Correlation analysis of $T_{well}$ in winter and summer regimes with: (a) the average daily pipe temperature $T_{pipe}$, (b) the average daily flow rate Q, (**c**) the average ambient temperature $T_{amb}$, (**d**) the daily cumulated solar radiation $\sum G_{i}$.

**Table 1 sensors-23-03121-t001:** Experimental results at laboratory conditions (P = 1 bar).

		Low flow (0.5 m s^−1^)					High flow (7 m s^−1^)				
		Winter Regime		Summer Regime			Winter Regime		Summer Regime		
*P* = 1 bar	$T_{set}$ (°C)	8	12	30	40	50	8	12	30	40	50
	$T_{flow}$ (°C)	20.06	20.14	24.26	25.80	28.20	20.94	20.71	22.89	23.62	24.05
	$T_{well,meas}$ (°C)	15.85	17.52	27.56	33.48	39.80	19.45	20.01	24.77	27.53	30.49
	$E_{meas}$ (°C)	−4.21	−2.62	3.30	7.68	11.60	−1.49	−0.70	1.88	3.91	6.45
	$E{\%}_{meas}$ (%)	−1.44%	−0.89%	1.11%	2.57%	3.85%	−0.51%	−0.24%	0.64%	1.32%	2.17%
	$T_{pipe,ext}$ (°C)	8.32	12.53	30.47	40.71	50.74	11.09	14.54	30.27	39.92	49.58
	$T_{pipe,int}$ (°C)	10.10	13.62	29.34	38.49	47.34	16.01	17.65	27.04	32.96	39.13
	$T_{flow,in}$ (°C)	20.87	20.93	23.02	22.51	22.97	21.11	20.88	22.84	23.44	23.68
	$T_{flow,out}$ (°C)	17.84	18.74	26.00	30.08	34.67	19.75	19.94	24.14	26.22	28.16

**Table 2 sensors-23-03121-t002:** Validation of the numerical model and estimation of the errors at different line pressure.

		Low Flow (0.5 m s^−1^)					High Flow (7 m s^−1^)				
		Winter Regime		Summer Regime			Winter Regime		Summer Regime		
$T_{set}$ (°C)		8	12	30	40	50	8	12	30	40	50
*P* = 1 bar	$T_{flow}$ (°C)	20.06	20.14	24.26	25.80	28.20	20.94	20.71	22.89	23.62	24.05
	$T_{well,meas}$ (°C)	15.85	17.52	27.56	33.48	39.80	19.45	20.01	24.77	27.53	30.49
	$T_{well,model}$ (°C)	12.44	15.14	28.21	35.73	43.26	18.97	19.48	24.60	27.52	30.45
	${∆}_{model}$ (°C)	−3.41	−2.38	0.65	2.25	3.46	−0.48	−0.53	−0.17	−0.01	−0.04
	$∆{\%}_{model}$ (%)	−1.16%	−0.81%	0.22%	0.75%	1.15%	−0.16%	−0.18%	−0.06%	0.00%	−0.01%
	$E_{model}$ (°C)	−7.62	−5.00	3.95	9.93	15.06	−1.97	−1.23	1.71	3.90	6.40
	$E{\%}_{model}$ (%)	−2.60%	−1.70%	1.33%	3.32%	5.00%	−0.67%	−0.42%	0.58%	1.31%	2.15%
*P* = 5 bar	$T_{well,model}$ (°C)	14.53	16.50	27.17	33.18	39.46	20.10	20.19	23.63	25.32	26.88
	$E_{model}$ (°C)	−5.53	−3.64	2.91	7.38	11.26	−0.84	−0.52	0.74	1.70	2.83
	$E{\%}_{model}$ (%)	−1.89%	−1.24%	0.98%	2.47%	3.74%	−0.29%	−0.18%	0.25%	0.57%	0.95%
*P* = 24 bar	$T_{well,model}$ (°C)	17.01	18.12	25.91	30.03	34.72	20.72	20.57	23.09	24.09	24.84
	$E_{model}$ (°C)	−3.05	−2.02	1.65	4.23	6.52	−0.22	−0.14	0.20	0.47	0.79
	$E{\%}_{model}$ (%)	−1.04%	−0.69%	0.55%	1.41%	2.16%	−0.07%	−0.05%	0.07%	0.16%	0.27%
*P* = 30 bar	$T_{well,model}$ (°C)	17.34	18.33	25.74	29.60	34.07	20.77	20.60	23.05	23.99	24.68
	$E_{model}$ (°C)	−2.72	−1.81	1.48	3.80	5.87	−0.17	−0.11	0.16	0.37	0.63
	$E{\%}_{model}$ (%)	−0.93%	−0.62%	0.50%	1.27%	1.95%	−0.06%	−0.04%	0.05%	0.12%	0.21%

## Data Availability

Not applicable.

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
