# Peer review of "On the Reliability of Temperature Measurements in Natural Gas Pipelines"

_sensors, 2023, doi:10.3390/s23063121_

Round 1
Reviewer 1 Report
1. The range of experimental temperature to reproducing the winter should be enriched, at least it should include the temperature below 0 ℃。
2. The studied flow velocity should be enriched. In practice, the gas flow velocity is always large than 7m/s, in particular for high pressure pipe.
3. The developed numerical model should be validated by more experimental data under different conditions.
Author Response
Please, see the attached file

Reviewer 2 Report
The paper titled “Reliability of temperature measurements in natural gas networks” is reviewed and the authors must revise the paper before it is going to be accepted.
1. The authors must discuss the most relevant literature from the reputed publications in introduction section.
2. Improve figures quality.
4. The conclusion must be revised with major findings and also incorporate the scope of this work in future.
5. References must be updated with the most recent articles last 5 years.
6. The paper must be minutely checked for language corrections.
Author Response
Please, see the attached file

Reviewer 3 Report
Comments:
1. The English expression of the manuscript needs to be improved. Pay attention to the sentence logic and avoid too long sentences.
For example, in Abstract “The results obtained in the laboratory experi-17 mental campaign and reported at the typical line pressure of 30 bar show errors ranging between 0.16°C and 5.87°C in summer regime and between -0.11°C and -2.72°C in winter regime, depending on the external pipe temperature (set at 8, 12, 30, 40 and 50 °C) and on the gas velocity (set at 0.5 and 7 m/s).”
2. In par.2.3 say that Tflow is not available. Use the model in par.2.2 to obtain Tflow and calculate the temperature error. Whether the theoretical model is applicable to the actual complex pipeline and whether the calculation results are reliable.
“Since a shielded sensor installed directly in the gas stream in the proximity of the thermowell was not available, the reference temperature of the gas stream, Tflow , was obtained through the model described in par. 2.2.”
3. “In this paper the authors present the results of a numerical and experimental study conducted both in the laboratory and on the field aimed at analyzing the reliability of temperature measurement in natural gas networks as a function of the pipe temperature and of the pressure and velocity of the gas stream.”
The purpose of the manuscript is to analyze the temperature reliability and the relationship between pipeline temperature、pressure、velocity, and obtain the difference between the temperature reliability in winter and summer. Whether there is further information exploration or innovative research.
4. The specific error of temperature reliability is obtained in the conclusion part of the manuscript, but the measurement error is inevitable, and whether the conclusions obtained are universal.
5. Most of the References overpass five years, and the format specification need to be strengthened.

Author Response
Please, see the attached file

Round 2
Reviewer 3 Report
I have no comments to authors.